# Disruption of Poly(ADP-ribosyl)ation Improves Plant Tolerance to Methyl Viologen-Mediated Oxidative Stress via Induction of ROS Scavenging Enzymes

**DOI:** 10.3390/ijms25179367

**Published:** 2024-08-29

**Authors:** Natalia O. Kalinina, Nadezhda Spechenkova, Irina Ilina, Viktoriya O. Samarskaya, Polina Bagdasarova, Sergey K. Zavriev, Andrew J. Love, Michael Taliansky

**Affiliations:** 1Shemyakin-Ovchinnikov Institute of Bioorganic Chemistry of the Russian Academy of Sciences, Moscow 117997, Russia; solanum@ibch.ru (N.S.); ilina@ibch.ru (I.I.); v.sam@ibch.ru (V.O.S.); polina44815283v@gmail.com (P.B.); szavriev@ibch.ru (S.K.Z.);; 2Belozersky Institute of Physico-Chemical Biology, Lomonosov Moscow State University, Moscow 119234, Russia; 3The James Hutton Institute, Invergowrie, Dundee DD2 5DA, UK; andrew.love@hutton.ac.uk

**Keywords:** poly(ADP-ribosyl)ation, poly(ADP-ribosyl) polymerase 1, oxidative stress, methyl viologen, salicylic acid

## Abstract

ADP-ribosylation (ADPRylation) is a mechanism which post-translationally modifies proteins in eukaryotes in order to regulate a broad range of biological processes including programmed cell death, cell signaling, DNA repair, and responses to biotic and abiotic stresses. Poly(ADP-ribosyl) polymerases (PARPs) play a key role in the process of ADPRylation, which modifies target proteins by attaching ADP-ribose molecules. Here, we investigated whether and how PARP1 and PARylation modulate responses of *Nicotiana benthamiana* plants to methyl viologen (MV)-induced oxidative stress. It was found that the burst of reactive oxygen species (ROS), cell death, and loss of tissue viability invoked by MV in *N. benthamiana* leaves was significantly delayed by both the RNA silencing of the *PARP1* gene and by applying the pharmacological inhibitor 3-aminobenzamide (3AB) to inhibit PARylation activity. This in turn reduced the accumulation of PARylated proteins and significantly increased the gene expression of major ROS scavenging enzymes including SOD (NbMnSOD; mitochondrial manganese SOD), CAT (NbCAT2), GR (NbGR), and APX (NbAPX5), and inhibited cell death. This mechanism may be part of a broader network that regulates plant sensitivity to oxidative stress through various genetically programmed pathways.

## 1. Introduction

ADP-ribosylation (ADPRylation) is a versatile reversible posttranslational protein modification system in eukaryotic cells which participates in the regulation of a broad range of biological processes, including DNA repair, programmed cell death, cell signaling, and responses to biotic and abiotic stresses [1]. Poly(ADP-ribosyl) polymerases (PARPs) play a key role in the process of ADPRylation, which modifies target proteins by attaching ADP-ribose molecules, which are provided by nicotinamide adenine dinucleotide (NAD+). This reaction leads to either decoration of the proteins with a single ADP-ribose moiety (MARylation) or with poly(ADP-ribose) (PAR) chains of differing lengths and branching architectures (PARylation) [1,2]. This also results in the formation of nicotinamide as a collateral product [1,3]. ADPRylated proteins, including PARP itself and other protein targets, act as modulators of various signaling networks. Once their functions have been completed, poly(ADP-ribose) glycohydrolases (PARGs) cleave the PAR chains linked to the proteins, which liberates free PAR or ADP-ribose [1]. Free PAR is then hydrolyzed into AMP and ribose-5-phosphate via the function of nucleoside diphosphate linked to some moiety-X (NUDIX) hydrolases, which are specific for ADP-ribose pyrophosphates or ADP-ribose [4].

In plants, the PARP family is represented by three proteins, PARP1, PARP2, and PARP3. Two of them, PARP1 and PARP2, are nuclear and have PARylation activities [1,5,6]. The functional role of plant PARP3 remains largely unelucidated. Considering the structural domains and architectures of these proteins, Arabidopsis PARP1 and PARP2 are similar to human HsPARP1 and HsPARP2, respectively; however, PARP3 has no human counterpart [1,5]. Arabidopsis also has two *PARG* genes (*PARG1* and *PARG2*) which encode products that have canonical PARG catalytic activity in vivo and in vitro [5]. Plants have a diverse variety of NUDIX hydrolases, six of which (NUDIX2, 6, 7, 10, 14, and 19) in Arabidopsis have pyrophosphohydrolase activities against ADP-ribose [7]. These can therefore also be considered as key components in the entire PARylation process. Plants, like animals, exploit PARylation processes to regulate multiple biological functions such as DNA damage responses, plant development, RNA biogenesis, plant immunity, and abiotic stress responses [1,2,4,8].

A key hallmark of abiotic stress is the accelerated production of reactive oxygen species (ROS), largely generated in plant mitochondria and chloroplasts, which function as centers of ROS synthesis, metabolic control, and also coordinate stress responses in other cell compartments [9]. ROS accumulation results in cellular oxidative damage [10] via protein oxidation, lipid peroxidation, DNA damage, reduction of photosynthetic pigment content, reduced photochemistry efficiency, and impaired photosystem II function. This damage may culminate in plant cell death [11,12].

Over the past decade, methyl viologen (MV; also known as the herbicide paraquat) has been used as an experimental tool (redox cycling oxidative stress inducer) to study general ROS responses and elucidate the basis of oxidative stress tolerance. Major ROS include the superoxide anion (O_2_^−^), hydrogen peroxide (H_2_O_2_), and hydroxyl radicals (OH) [9,13]. In plants, multiple mechanisms have been proposed to account for MV resistance including ROS detoxification by enzymatic antioxidants (scavenging enzymes) [14,15], MV transport or sequestration [16], highly active reductive metabolism, energy salvaging pathways, redox transfer between cellular compartments [17], activation of phytohormone (abscisic acid, ABA or salicylic acid, SA) signaling pathways [8,9], and suppression of plant caspase-like (phytaspase) activities [18].

Given the central role of PARP1 in regulating oxidative stress responses, it is important to explore how its reduced activity might contribute to MV resistance in plants [19,20,21]. However, it is far from clear how the resistance to MV-induced oxidative stress is regulated by PARP. Interestingly, the Arabidopsis *rcd1* (*radical-induced cell death1*) mutant exhibits a high level of resistance to MV-induced damage [17]. RCD1 is a PARP-like protein belonging to the SRO-gene family (Similar-to-RCD1), yet it does not have direct PARP activity [22]. MV tolerance in *rcd1* has been demonstrated to be directly associated with the effect on reductive metabolism and rerouting of the energy production pathways [17] but apparently not with PARP-mediated ribosylation activity.

We undertook this study to investigate how PARP1 and PARylation regulate the responses of *Nicotiana benthamiana* plants to MV-induced oxidative stress. We showed that the cell death, ROS burst, and loss of viability induced by MV in *N. benthamiana* leaves was significantly delayed by RNA silencing of the *PARP1* gene and by application of a pharmacological inhibitor of PARylation activity. The reduction in PARylated proteins was associated with a significant increase in the expression of major ROS scavenging enzymes, such as CAT (NbCAT2), SOD (NbMnSOD), APX (NbAPX5), and GR (NbGR), which contributed to the inhibition of cell death. This mechanism may be integrated in a specific consolidated network that controls plant sensitivity to oxidative stress by multiple genetically programmed pathways.

## 2. Results

### 2.1. RNAi Silencing as Well as Pharmacological Inhibition of PARP1 Induces Tolerance to MV—Mediated Stress in Nicotiana Benthamiana

To examine a possible link between the cell death-related responses to oxidative stress and PARP1 activities, we first used MV as a redox cycling oxidative stress inducer [23] and confirmed that treatment of *N. benthamiana* leaf discs with MV resulted in the loss of their viability (bleaching) [18] (Figure 1A,B). This effect was associated with an important hallmark of programmed cell death, the accumulation of ROS [24], which was expressed as H_2_O_2_ production (Figure 1D,E) and which is a consistent and reliable marker of ROS production [9,18]. This suggests that MV induces a ROS burst. Then we used a virus-induced gene silencing (VIGS) approach to knock down *PARP1* expression. VIGS has been proven to be a robust, fast, and efficient method to study functions in plant growth and development, cellular metabolic and signaling pathways, and responses to various biotic and abiotic stresses [25]. A potato virus X (PVX) vector has been successfully developed and exploited in a variety of loss-of-function experiments in *N. benthamiana* plants [25]. VIGS constructs were produced by inserting two PARP1 gene fragments into a PVX vectorpGR106 [26], creating PVX-PARP1 (I) and PVX-PARP1 (II). Expression levels of PARP1 mRNA were suppressed approximately ten-fold in both cases by PVX-PARP1, independent on the presence or absence of MV, when compared with an empty PVX vector (PVX) used as a negative control (Figure 1C). Moreover, silencing in knock-down (KD) VIGS lines considerably enhanced the viability (suppressed bleaching) of leaf discs and impaired H_2_O_2_ accumulation in comparison to control plants (Figure 1A,D).

To confirm our findings obtained in genetic experiments using VIGS to silence the *PARP1* gene, we used an additional pharmacological approach based on the effect of the PARP inhibitor 3-aminobenzamide (3AB), which targets the PARP conserved enzymatic active site [26]. We found that the treatment of *N. benthamiana* leaf discs with 3AB substantially increased resistance to MV in terms of cell viability and prevented a MV-induced ROS burst (Figure 1D,E).

Collectively, these data support the idea that during oxidative stress, PARP1 may facilitate ROS-mediated cell death, and disruption of its expression or activity in *N. benthamiana* enhances stress tolerance.

### 2.2. MV Does Not Affect Gene Expression and Subcellular Localization of PARP1

To examine a possible link between PARP1 activities and MV-mediated cell death in *N. benthamiana*, we analyzed the effect of MV on *PARP1* gene expression by an RT-qPCR assay and found that PARP1 mRNA levels were not significantly affected by MV in *N. benthamina* plants compared with control MV-untreated plants (Figure 1B). Moreover, no noteworthy differences were observed in PARP1 mRNA levels in PVX (empty vector)-infected plants caused by MV treatment (Figure 1B). Previously, we obtained similar results showing no effect of virus (tobacco rattle virus, TRV) infection on the expression of the *PARP1* gene. However, we observed that TRV induced the intracellular redistribution of PARP1 that led to the activation of PARP1 activity and consequent antivirus defense [26]. Therefore, in the next series of experiments, we examined the subcellular localization of PARP1 in *N. benthamiana* plants.

PARP1 is a nuclear enzyme, and its activities are intimately tied to nuclear compartments. Of note are the interactions of PARP1 with the nucleolus and Cajal bodies (CBs). The nucleolus and CBs are the prominent subnuclear domains that have traditionally been implicated in various RNA-related processes [27]. PARP1 molecules that are unmodified normally accumulate in the nucleolus and associate with chromatin. However, PARylation automodified PARP1 and other additional PARylated proteins can associate with coilin, the key structural protein of CBs [28]. This interaction likely controls the movement of PARP1 and other PARylated proteins from the nucleolus and chromatin into CBs for PAR cleavage and recycling by PARG. Therefore, as expected, using a fluorescently labelled anti-PARP1 antibody, we found that in the absence of MV treatment, PARP1 was located in the nuclei and nucleoplasm (presumably binding to chromatin), preferentially targeting CBs (Figure 2). It is noteworthy that MV treatment did not affect this intracellular localization of PARP1 at all.

### 2.3. MV Activates Poly(ADP-ribosyl)ation

Given that silencing (VIGS) of the *PARP1* gene enhances tolerance to MV-mediated stress (Figure 1), it could be expected that in spite of the absence of functional effects of MV on *PARP1* gene expression and cellular localization, MV may still increase levels of PAR associated with PARP1 target proteins. Indeed, we showed that PAR accumulated to significantly higher levels in unsilenced (mock or PVX infected) *N. benthamiana* leaf discs treated with MV compared with untreated unsilenced plants (Figure 3A,B). Thus, the over-accumulation of PARylated proteins may be the result of post-transcriptional activation of PARP1 rather than transcriptional regulation of *PARP1* expression. Indeed, it is known that PARP1 activity may be subjected to various PARP1 modifications such as binding to DNA breaks, methylation, phosphorylation, or its interaction with some proteins and small nucleolar RNAs (snoRNAs), which increase PARP1 catalytic activity. PARP1 binds to snoRNAs, which stimulate PARP1 catalytic activity [30,31].

However, silencing (VIGS) of the *PARP1* gene led to a significant reduction in PAR levels in *N. benthamiana* leaves treated with MV (Figure 3A), which was accompanied by an increase in tolerance to MV. These findings obtained by the VIGS studies were confirmed in the experiments using the 3AB PARP1 inhibitor, which showed that reduction of PAR accumulation (Figure 3B) is associated with enhanced resistance to MV-mediated stress (Figure 1A,B).

Thus, the pharmacological (3AB inhibitor) and genetic (VIGS) studies have revealed a negative correlation between over-accumulation of PARylated proteins and the plant responses which facilitate tolerance to MV, suggesting a direct interplay between PARylation and plant tolerance to oxidative stress.

### 2.4. The Role of PARP1 in Accumulation of Plant Hormones

Given that antivirus defence induced by PARP1 activity (PARylation) was attributed to its ability to facilitate the generation of endogenous SA and activate SA-mediated signaling pathways [24], we investigated whether and how PARP1 may interact with SA and another plant hormone, ABA, during MV-mediated oxidative stress.

We found that in the absence of MV, neither PARP1 silencing (VIGS) nor its activity inhibition (3AB) affected the comparatively low levels of both free SA and conjugated SA (SA-β-glucoside, SAG) [32] (Figure 4A,B). However, in the presence of MV, the *PARP1* silencing or 3AB-mediated pharmacological inhibition of PARP1 activity and consequent decrease in the amount of PARylated proteins during MV-mediated stress coincided with a significant accumulation of free SA and SAG (Figure 4A,B). Interestingly, a reduction in the accumulation of PARylated proteins during TRV infection positively correlated with a decrease of free SA and SAG concentrations [26]. However, why PARylation differentially modulates the accumulation of SA during MV treatment and TRV infection remains unclear.

ABA is another phytohormone that plays an important role in responses to abiotic stresses [33]. The accumulation of ABA by PARylation under MV-mediated stress was modulated in the same manner as that of SA (Figure 4C).

Collectively, these data suggest that the mechanism of plant responses to MV-mediated stress involves a functional interplay between PARP-triggered PARylation and plant hormone metabolism and signaling.

### 2.5. RNAi Silencing and Pharmacological Inhibition of PARP1 Activates Expression of Genes Encoding Major ROS Scavenging Antioxidant Enzymes

Plant cells have several protective mechanisms to manage oxidative stress, most of which involve production of cytoprotective antioxidant enzymes that scavenge ROS and are crucial for stress tolerance in plants. Antioxidant enzymes are key players in the conversion of ROS and their derivatives into stable non-toxic molecules; these represent a crucial defense mechanism against oxidative stress-induced cellular damage. The hazardous effects of oxidative stress and ROS are countered by enzymes belonging to catalase (CAT), superoxide dismutase (SOD), ascorbate peroxidase (APX), and glutathione reductase (GR) antioxidant protein families [34,35,36,37]. Therefore, we examined whether the gene expression of members of these families was modulated by the disruption of PARylation activity caused by the silencing (VIGS) of the *PARP1* gene or PARP1 pharmacological inhibition with 3AB. Our results show that both genetic (VIGS) and pharmacological suppression of MV-mediated PARylation activity is accompanied by increased transcription rates of genes encoding CAT (*NbCAT2*), SOD (*NbMnSOD*, mitochondrial manganese SOD), APX (*NbAPX5*), and GR (*NbGR*) (Figure 5A–D), thereby increasing the antioxidant activities of cells and consequently enhancing their tolerance to MV-mediated oxidative stress.

To validate the RT-PCR data, the enzymatic activities of two ROS scavenging enzymes (SOD and CAT) were measured using colorimetric activity kits. The activity trends of these two enzymes were highly consistent with those from gene expression analyses by RT-qPCR (Figure 6).

## 3. Discussion

Plants are constantly exposed to abiotic stresses, both physiological and environmental, that can negatively impact their metabolism, growth, and development. These negative impacts are likely to be further exacerbated as a consequence of future global climatic changes, which will increase the risk of physiological harm. Over the past decade, researchers have employed various approaches to uncover the molecular mechanisms underlying plant stress responses and to identify the key genes that regulate these processes [38,39]. Many studies have implicated oxidative stress as a major component of plant abiotic stress perception and downstream signaling cascades [40,41]. Oxidative stress is a complex physiological state caused by an imbalance between the production of ROS (highly damaging to cells) and the ameliorative antioxidant activity that accompanies virtually all biotic and abiotic stresses in plants [41]. The fine balance between ROS production and antioxidant activity strongly influences plant survival under stress and can often have irreversible effects on tissue and organ development, which may lead to abnormal plant growth or death [42]. These effects can be brought about directly by the activity of ROS and their capacity to oxidize and attack cellular components, and also via their interplay with epigenetic modifiers and hormones which can further influence plant developmental processes and stress responses [43]. The effect of ROS on plant physiology depends on their different levels of reactivity, magnitude and sites of production, and potential to cross biological membranes [43].

In addition to abiotic stresses, ROS play a pivotal role in plant defense against pathogens and have been implicated in HR-associated programmed cell death and systemic acquired resistance during pathogen attack [44], activating, for example, caspase-like proteins [18]. Thus, the mechanisms influencing plant resistance and susceptibility to pleiotropic stresses cannot depend on a single devoted regulatory component; rather, it is likely composed of overlapping regulatory networks which orchestrate a specific suite of responses to various biotic and abiotic stresses.

There are multiple lines of convincing evidence that PARylation plays a key role in plant responses to different stresses. RNA interference-mediated silencing of *PARP* genes in oilseed rape and Arabidopsis, which enhanced the plant tolerance to drought, heat, MV treatment, and high light, also resulted in decreased PAR accumulation and PARP activity [1,4,19]. However, the underpinning mechanisms of such tolerance remain unelucidated. One possibility is that silencing PARP prevents the excessive energy consumption typically associated with its activation under stress conditions [19]. Another possibility is that PARP silencing could result in an increase of abscisic acid (ABA) levels and concomitant activation of ABA-responsive genes [19]. ABA is a major plant hormone which is involved in plant responses to abiotic stress [45]. The data implied that PARP proteins or PARylation may influence abiotic stress responses as an ABA-dependent negative regulator. Consistent with the data, it was found that an Arabidopsis *parg* knockout mutant demonstrated increased sensitivity to oxidative stress; thus, increased PARylation rates are likely needed for such sensitivity [46].

In this paper, we studied the role of PARylation in plant responses to MV-mediated oxidative stress. PARylation is catalyzed by members of the PARP family. Plants encode three PARP proteins, but only two of them (PARP1 and PARP 2, but not PARP3) possess poly(ADP-ribose) polymerase activity [6]. PARP2 is regarded as the predominant PARP enzyme in plant DNA damage and immune responses [47]. However, in our previous study, we found that there is a PARP1 that plays a key role in the regulation of host defenses against a virus [26]. Therefore, here, we extend our previous research by elucidating the role of PARP1 in response to MV-mediated abiotic stress. The role of the entire PARP family will be investigated in the future. We demonstrate that RNAi silencing (VIGS) as well as pharmaceutical inhibition (3AB) of PARP1 induces tolerance to MV-mediated stress in *N. benthamiana*, decreasing overaccumulation of ROS and preventing cell death. It is still unclear if PARP2 can also contribute to this process, and this will be investigated in future. These events are accompanied by a significant reduction of PARylation rates increased by MV alone. These findings suggest that excessive PARylation may harm plant cells under MV-mediated stress. This is in clear contrast to the previous findings that exaggerated PARylation levels facilitate cell viability and plant defense during biotic stress (virus infections) [26,48]. Interestingly, neither virus infection (TRV) [26] nor MV treatment (Figure 2) were able to modulate *PARP1* gene expression, which suggests other regulatory mechanisms, likely operating on the level of activity/location of PARP1 protein, could be involved. In the case of TRV, the CB protein coilin interacts with PARP1 and redistributes it and retains it at the nucleolus, preventing its trafficking from the nucleolus to CBs for PAR cleavage and recycling. These events are accompanied by substantial increases in endogenous concentrations of the plant hormone salicylic acid, leading to the restriction of TRV invasion [26]. Given that 3AB directly inhibits PARP1 activity, it is reasonable to suggest that the PARP1 enzyme is involved in the MV-mediated overaccumulation of PARylated proteins; however, the potential role of PAR-degrading enzymes like PARG and NUDIX, whether through deactivation or suppression of gene expression, cannot be entirely excluded. Moreover, since MV does not modify *PARP1* gene expression, it is conceivable that the enhanced accumulation of PARylated proteins is caused by activation at the posttranscriptional level, but how exactly is not known yet. One possibility is that PARP1 can be activated by single- and double-strand DNA breaks as described for some animal systems [49]. However, other abiotic stress-induced factors may also be involved in plants.

Thus, these RNAi silencing and pharmacological (3AB) experiments strongly support the idea that PARylation is involved in cell death pathways induced by MV-mediated oxidative stress operating upstream of ROS accumulation.

Another interesting observation is that the decreased rates of PARylation caused by *PARP1* silencing or chemical inhibition correlate well with the tolerance to MV-mediated stress, which presumably was achieved by the transcriptional activation of genes encoding ROS scavenging enzymes such as CAT, SOD, APX, and GR.

We propose a model demonstrating the integral connection between poly(ADP-ribosyl)ation activity mediated by PARP1 in response to oxidative stress caused by MV, changes in production of the phytohormones SA and ABA, induction of ROS scavenging enzymes, and stress tolerance. In this model, MV treatment of plants post-transcriptionally activates PARP1 which in turn causes the overproduction of PARylated proteins. The suppression of this process by RNA silencing of the *PARP1* gene or pharmacological (3AB) inhibition positively correlates with the enhanced production of SA and ABA, which consequently activate ROS scavenging enzymes, conferring tolerance to MV-mediated stress.

Our data show that protein PARylation plays an important role in plant responses to abiotic stress. Importantly, MV treatment enhances PARylation activity, suggesting that PARylation is an integral part of the regulatory network of stress tolerance in plants. However, PARylation targets and their functions remain largely unknown in plants. We cannot exclude that excessive PAR accumulation induced by MV on its own may merely cause cell death, similar to some mammalian systems [50], though more specific protein targeting is also possible. For example, taking into account the cross talk between PARylation and SA/ABA, we could hypothesize that the proteins involved in SA/ABA biosynthesis or signaling may serve as substrates for PARylation. On a practical level, fine tune control of PARylation may constitute a powerful technique to modulate crop plant stress responses. However, such approaches should be carefully considered, given that the sensitivity/tolerance to different stresses may be controlled by different degrees of PARylation. Long-term effects of PARP1 inhibition on plant growth and development might also have unintended consequences, such as affecting DNA repair processes or other vital cellular functions, which could compromise plant health. That is why genetic technologies (such as CRISPR-Cas) leading to permanent irreversible *PARP1* gene modifications may not be suitable in agricultural practices. Spray-induced genetic silencing, using the application of dsRNA, may be a more applicable approach.

## 4. Materials and Methods

### 4.1. Virus-Induced Silencing of PARP1 Expression

Two fragments of NbPARP1 were amplified as described by Spechenkova et al., 2023 [26] using primers GAGTGCTCCAAAAAGCATCC and TGGATGGGATAGCCTCTCAG to produce a 440 nt fragment 1 from nucleotides 491 to 968. Primers CCGCTTATAATTAAACCTCAC and GGACTAAGAATTGCTCCTCCA were used to produce a 410 nt fragment 2 at the 3′ terminus of the *NbPARP1* gene. These were separately spliced in an antisense orientation into the PVX vector (pGR106) genome [51] to produce two different NbPARP1-silencing constructs (PVX-PARP). The negative control consisted of an empty PVX vector (PVX). Three lower leaves of four *N. benthamiana* plants at the 4–5 leaf stage of development were infiltrated with *A. tumefaciens* GV3101 cultures (OD_595_ = 0.1) carrying PVX-PARP1 VIGS or PVX control constructs. An additional control, namely plants untreated with *Agrobacterium* (-PVX), was also used. Five days post- agroinfiltration, plants were inoculated with TRV. After ten days, samples of leaves were collected from either the apical tip (systemically infected) or inoculated. The four leaf samples from plants with each respective treatment were pooled prior to RNA extraction and analysis as discussed above. Two separate PVX-NbPARP1 VIGS constructs which were made in this work exhibited similar effects on *PARP1* gene expression.

### 4.2. MV Stress and 3AB Treatments

In order to examine the effect of *PARP1* silencing of MV stress responses, leaf discs were cut out of the VIGS-silenced (PVX-PARP) and unsilenced control (PVX and mock-inoculated) *N. benthamiana* plants 9 days post infection, immersed in an aqueous solution of MV at a concentration of 10 µM or in a control solution (water), vacuum infiltrated, and incubated for up to 96 h in continuous light in 96-well plates as described by Chichkova et al., 2010 [18]. In order to examine the effect of the PARP inhibitor 3AB, leaf discs were cut out of the uninfected healthy *N. benthamiana* plants, treated with MV as described above, and floated in 0.5% dimethyl sulfoxide (DMSO) with (+3AB) or without (−3AB) 2.5 mM 3AB (Sigma Aldrich; St. Louis, MO, USA) as described by Adams-Phillips et al. [52]. Then, leaf discs were photographed 24, 48, and 96 h after treatment.

### 4.3. Chemiluminescence Assay for H_2_O_2_

Generation of ROS in leaf discs was assayed 24 h after treatments described above by measuring the H_2_O_2_-dependent luminescence of luminol as described [18,53] using a scintillation spectrometer LS 6000SE (Beckman Coulter; Brea, CA, USA).

### 4.4. Immunolabelling and Confocal Imaging Analysis

*N. benthamiana* plant leaf tissues were fixed in 5% (*v/v*) dimethyl sulfoxide and 3.7% (*v/v*) formaldehyde in PHEM buffer (60 mM PIPES, 25 mM HEPES, 5 mM EGTA, 2 mM MgCl_2_, pH 6.9) for 2 h. Postfixation, the samples were digested with cell wall-degrading enzymes (1% cellulase, 0.1% pectolyase, and 0.1% bovine serum albumin in PHEM buffer) for 2 h. These were subsequently incubated for 20 min in 1% (*v/v*) Triton X-100 and treated with ice-cold methanol for 10 min. The samples were subsequently thrice washed with PHEM after each step. Localization of PARP1 was achieved via immunofluorescence approaches [29,54], which used primary rabbit antibodies to a KLH-conjugated synthetic peptide derived from Arabidopsis thaliana PARP1 (1:100; Agrisera; Vännäs, Sweden). Application of an Alexa Fluor 488-conjugated anti-rabbit secondary antibody (1:500; Invitrogen; Waltham, MA, USA) permitted visualization of the primary rabbit antibodies and thus indicated PARP1 location. Subcellular localization of the proteins was detected using a TCS SP2 confocal laser scanning microscope (Leica Microsystems; Deerfield, IL, USA). The acquired images were processed via Photoshop CC software ver. 19.0 (Adobe Inc; San Jose, CA, USA). To locate nuclei, the leaf tissues were infiltrated with PBS containing 4,6-diamidino-2-phenylindole (DAPI). CBs and nucleolar structures were detected by using a marker (fibrillarin fused to mRFP) which was delivered into cells via *Agrobacterium*-mediated expression [29].

### 4.5. Real Time Quantitative RT-PCR (RT-qPCR)

One to two grams of leaf tissue was ground to a fine powder in liquid nitrogen using a pestle and mortar, and RNA was subsequently extracted with TRI REAGENT (Sigma Aldrich) following the manufacturer’s protocols. RNA was suspended in 50 µL DEPC-treated water and resolved using electrophoresis in 1% agarose gel. This was subsequently electroblotted to Hybond N membrane before UV cross-linkage in a StrataLinker (Stratagene; La Jolla, CA, USA). RNase-free DNase I (Invitrogen) was used to remove any remaining DNA in the RNA sample. Aliquots of the treated RNA were used in transcription reactions (following the SuperScriptTM First-Strand Synthesis System for RT-PCR (Invitrogen), in conjunction with oligo-dT primer) to produce cDNA. The primer pairs for SYBR green-based real-time PCR analysis of APX (*NbAPX5*), PARP1 (*NbPARP1*), SOD (*NbMnSOD*), CAT (*NbCAT2*), and GR (*NbGR*) (designed using PRIMER EXPRESS software ver. 2.0; Applied Biosystems; Foster City, CA, USA) are listed in Appendix A. Concentrations of primers with the lowest threshold cycle (Ct) values were used in subsequent analysis in conjunction with 10-fold dilutions in sterile water of the first-strand cDNA reaction mixes. These components were combined with reagents in the QuantiTectTM SYBR^®^ Green PCR kit (Qiagen; Crawley, UK) according to the manufacturer’s suggestions, prior to placing in an ABI PRISM 7700 Sequence Detection System (Applied Biosystems) for amplification of products using the following reaction conditions: 95 °C for 15 min, followed by 40 cycles of 94 °C for 15 s, 60 °C for 30 s, and 72 °C for 30 s. The Ct value for each mRNA was normalized to reference gene mRNAs encoding Ubiquitin3 (UBI3) and the ribosomal protein L23 (L23) (Appendix A).

### 4.6. Immunological Detection of Poly ADP-Ribose (PAR)

For protein extraction, a Plant Total Protein Extraction Kit (Sigma Aldrich) was used. The protein was analyzed for PAR accumulation levels by ELISA using LysA™ Universal PARylation Assay Kit (BPS Bioscience; San Diego, CA, USA).

### 4.7. Measurement of Endogenous SA

Free SA and SAG were determined as described [26,55]. Leaf extracts in 70% aqueous EtOH (*v*/*v*) and o-anisic acid (OAA; as internal control) were centrifuged, and the supernatant was placed to another centrifuge tube. The pellet was resuspended in 90% MeOH and centrifuged. Both supernatants were combined, and the alcohol (EtOH and MeOH) was evaporated. The remaining solution was mixed with ethyl acetate and cyclohexane and centrifuged. The organic phase contained free SA; the aqueous phase contained SAG. The SAG-containing phase was diluted with 8 M HCl and heated for 1 h at 80 °C for hydrolysis of SAG. SA and SAG hydrolysate were quantified using liquid chromatography with fluorometric detection.

ABA was determined using a Plant hormone abscisic acid (ABA) ELISA Kit (Cusabio; Houston, TX, USA).

### 4.8. Quantification of SOD and CAT Activities

Measurement of SOD and CAT enzymatic activities was carried out using a Superoxide Dismutase (SOD) Colorimetric Activity Kit (Invitrogen) and a Catalase Colorimetric Activity Kit (Invitrogen).

## Figures and Tables

**Figure 1 ijms-25-09367-f001:**
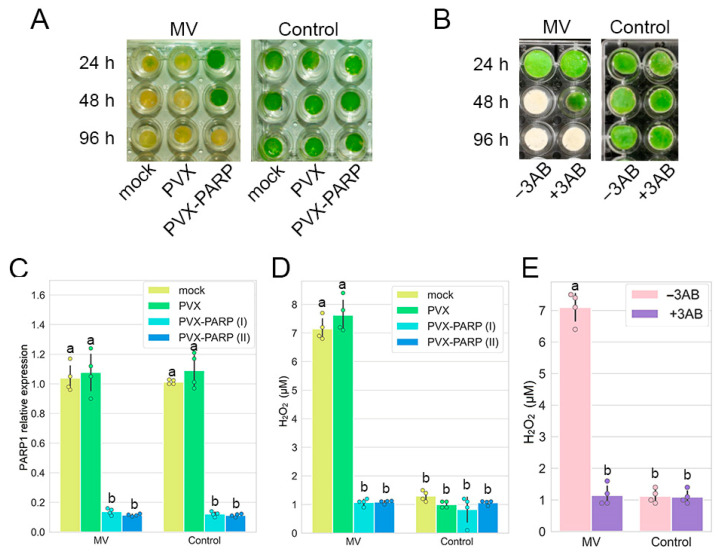
Effects of PARP1 deficiency on cell death in *N. benthamiana* plants induced by methyl viologen (MV). Effect of virus-induced silencing of *PARP1* (*NbPARP1*) (**A**,**C**,**D**) and the PARP inhibitor 3-aminobenzamide (3AB) (**B**,**E**) on loss of viability (bleaching) of leaf discs (**A**,**B**), expression of *PARP1* (**C**) and generation of H_2_O_2_ induced by MV (versus controls) (**D**,**E**). Two separate PVX-PARP1 VIGS constructs [PVX-PARP (I) and PVX-PARP (II), made as described previously [26], see Materials and Methods) exhibited similar effects on loss of viability and H_2_O_2_ generation. (**A**,**B**) Leaf discs were bathed in MV (10 µM) or control solutions (water) and photographed at 24, 48, and 96 h after treatment. (**C**) Level of virus-induced silencing of the *PARP1* gene in *N. benthamiana* facilitated by a PVX vector containing fragments of the *NbPARP1* gene [PVX-PARP (I) and PVX-PARP (II)] compared to an empty PVX vector control (PVX). PARP1 mRNA accumulation was quantified using RT-qPCR in systemically infected leaves at 10 days post-inoculation (dpi). Accumulation of PARP1 mRNA was measured by RT-qPCR and this was normalized against the internal *N. benthamiana* house-keeping controls, *60S ribosomal protein 23* gene (*L23*) and *UBIQUITIN3* gene (*UBI3*). Statistical analysis was performed on four independent biological replicates, where each replicate sample was derived from two leaves harvested from each plant of a group of three and pooled together. ANOVA and Tukey’s HSD post hoc analysis were performed on the RT-qPCR data. The different letters (a, b) denote significant differences in *p*-values (*p* < 0.001) of the PARP1 mRNA accumulation. (**D,E**) H_2_O_2_ accumulation was quantitated 24 h after treatment; data are means ± s.d. from three experiments with three independent replicates in each. ANOVA and Tukey’s HSD post hoc tests were performed on four independent biological replicates, where each replicate sample was derived from two leaves harvested from each plant of a group of three and pooled together. The different letters (a, b) indicate significantly different values (*p* < 0.001) in H_2_O_2_ accumulation.

**Figure 2 ijms-25-09367-f002:**
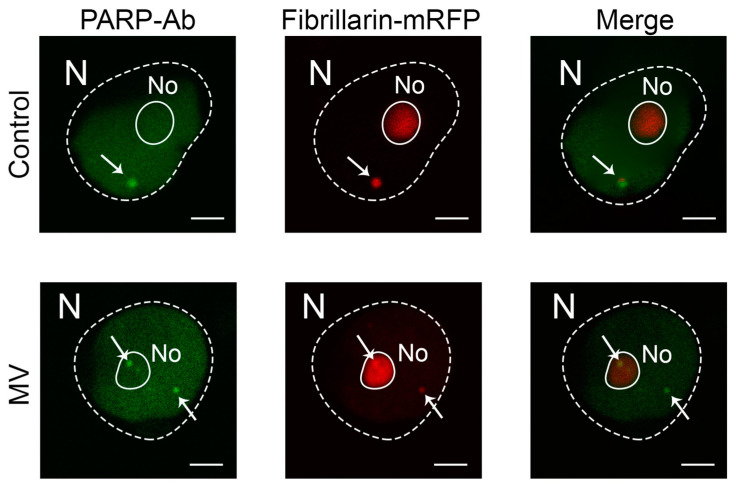
Representative images of the of PARP1 intranuclear localization in leaf discs of *N. benthamiana* plants in the presence or absence of MV (10 µM; 24 h after treatment), detected by immunofluorescent staining using primary rabbit anti-PARP1 antibody and secondary fluorescent anti-rabbit antibody (green). Ectopic *Agrobacterium*-mediated expression of fibrillarin, a nucleolar and CB marker [29]) tagged with monomeric red fluorescent protein (mRFP; magenta), was used to visualize nucleoli and CBs in the leaves. Merge images are presented on the right. N, nuclei, No, nucleoli, CBs, Cajal bodies (shown by arrows). Scale bars, 5 μm.

**Figure 3 ijms-25-09367-f003:**
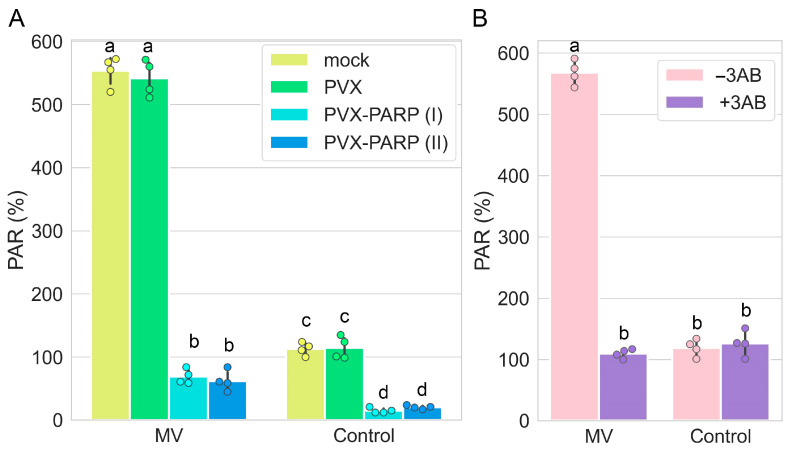
Accumulation of PARylated proteins induced by MV in leaf discs of VIGS-silenced *PARP1* (**A**) and 3AB-treated plants (**B**) 24 h after MV treatment. (**A**) Two separate PVX-PARP1 VIGS constructs (PVX-PARPI and PVX-PARPII), made as described previously [26], exhibited similar effects on PAR accumulation. Virus-induced silencing of the *PARP1* gene in *N. benthamiana* mediated by PVX-PARP constructs was compared with an empty PVX vector control (PVX). (**B**) Leaf discs were immersed in the solution of MV (10 µM) or control solution (lacking MV). Accumulation of PARylated proteins was measured by ELISA using rabbit anti-PAR polyclonal antibody. Statistical analysis was performed on four independent biological replicates. Replicates were derived from samples from two leaves per plant pooled from three plants. RT-qPCR data were statistically analyzed using Tukey’s HSD post hoc test and ANOVA. The different letters (a, b, c, d) indicate significant differences in PAR accumulation levels.

**Figure 4 ijms-25-09367-f004:**
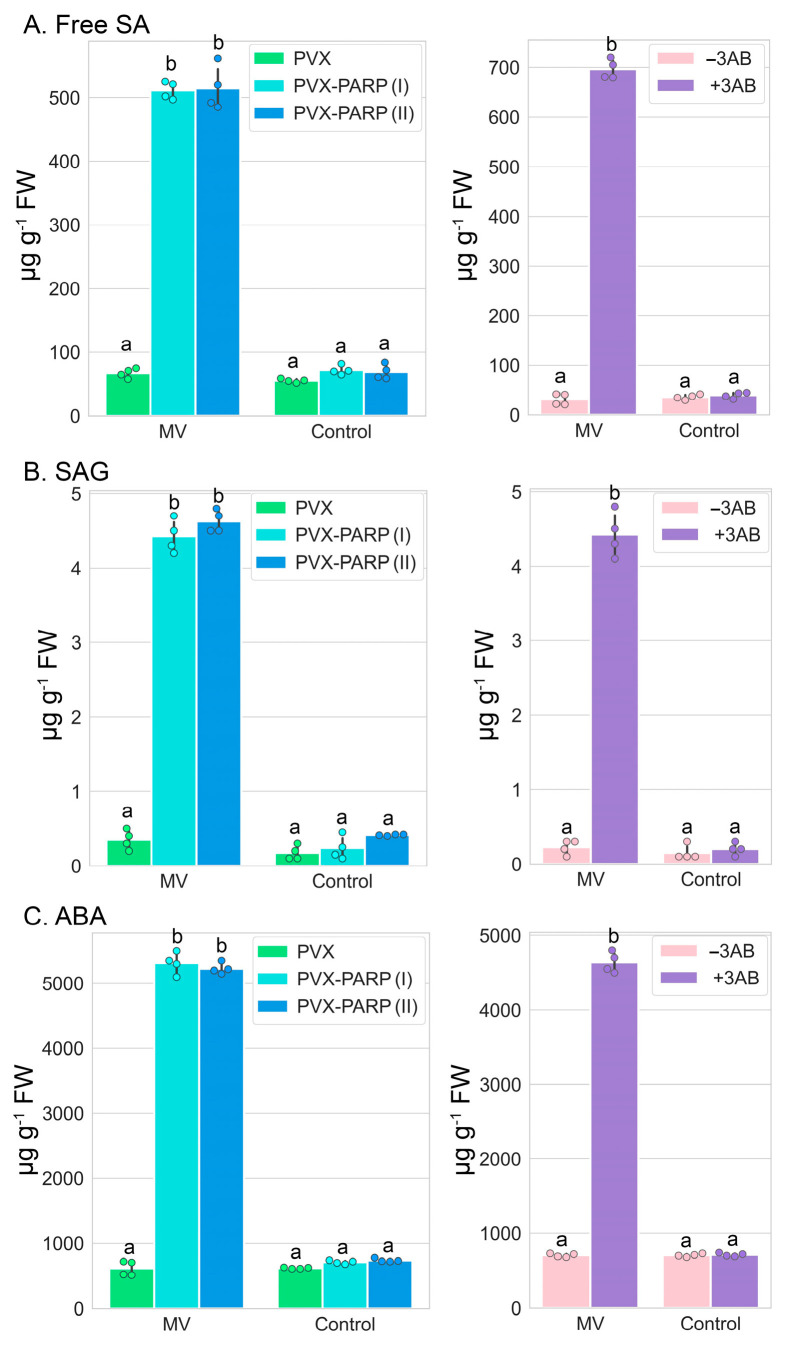
Concentrations of free SA (**A**), SAG (**B**), and ABA (**C**) in leaf discs of VIGS-silenced *PARP1* (**A**) and 3AB-treated plants (**B**) 24 h after MV treatment. Two separate PVX-PARP1 VIGS constructs (PVX-PARPI and PVX-PARPII), made as described previously [26], exhibited similar effects on accumulations of the phytohormones. Replicates were derived from samples from two leaves per plant pooled from three plants. RT-qPCR data were statistically analyzed using Tukey’s HSD post hoc test and ANOVA. The different letters (a, b) indicate significant differences in PAR accumulation levels.

**Figure 5 ijms-25-09367-f005:**
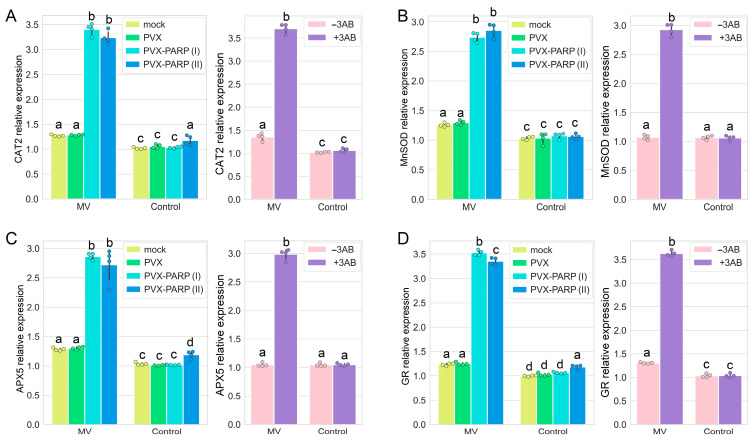
Modulation of ROS scavenging responses in *N. benthamiana* leaf discs by MV treatment (10 µM 24 h after treatment) via application of PARP1 VIGS or treatment with 3AB. (**A**–**D**) Expression patterns of ROS scavenging genes, encoding CAT (NbCAT2; **A**), SOD (NbMnSOD, mitochondrial manganese SOD; **B**), APX (NbAPX5; **C**), and GR (NbGR; **D**), were analyzed by RT-qPCR. mRNA levels of these genes were normalized to those of *UBIQUITIN3 gene (UBI3)* and *60S ribosomal protein 23 gene (L23)*, which act as housekeeping genes. Four independent biological replicates were each composed of samples derived from two leaves taken from each of three plants, pooled together. The obtained RT-qPCR data were analyzed using Tukey’s HSD post hoc test and ANOVA, whereby the different letters (a, b, c, d) indicate significantly different values (*p* < 0.001).

**Figure 6 ijms-25-09367-f006:**
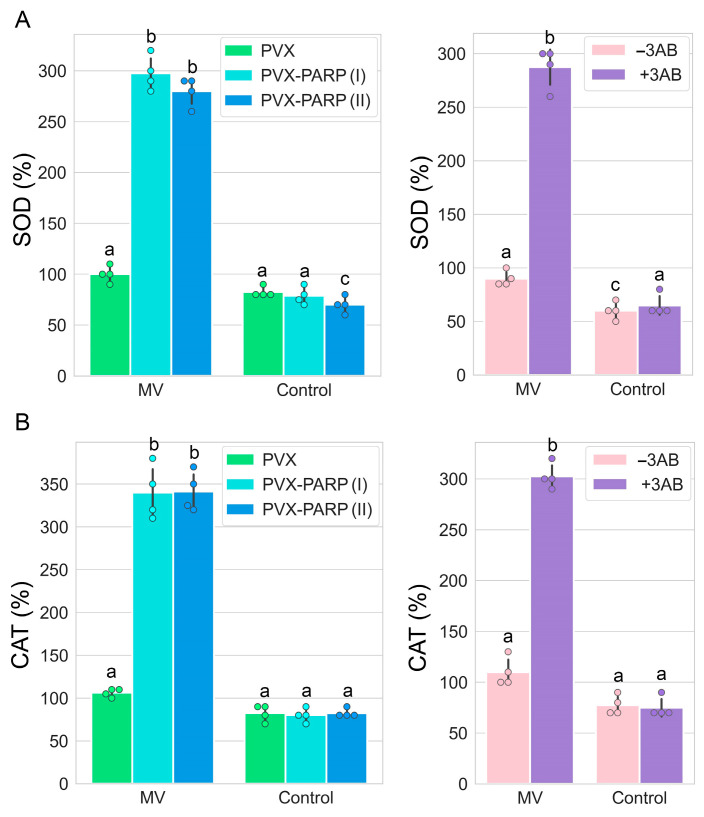
Modulation of activities of SOD (**A**) and CAT (**B**) ROS scavenging enzymes in *N. benthamiana* leaf discs by MV treatment (10 µM 24 h after treatment) via application of PARP1 VIGS or treatment with 3AB. Enzymatic activities were measured using Superoxide Dismutase (SOD) Colorimetric Activity Kit and Catalase Colorimetric Activity Kit, respectively. The obtained data were analyzed using Tukey’s HSD post hoc test and ANOVA, whereby the different letters (a, b, c) indicate significantly different values (*p* < 0.001).

## Data Availability

The NCBI GenBank accession number for the PARP1 gene reported in this paper is KP771975.

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
