# Peer review of "Disruption of Poly(ADP-ribosyl)ation Improves Plant Tolerance to Methyl Viologen-Mediated Oxidative Stress via Induction of ROS Scavenging Enzymes"

_ijms, 2024, doi:10.3390/ijms25179367_

Round 1

Reviewer 1 Report

Comments and Suggestions for Authors

This manuscript addressed the function of PARP1 in MV treatment through genetic (VIGS) and its pharmacological inhibitor (3AB). The data showed that VIGS plants exhibited resistance to MV, similar to addition of 3AB. Further analysis revealed that PARP1 can be PARylated to reduce ROS and conduct the resistant to MV treatment, which was associated with CAT, SOD, APX, and GR. This research is interesting, but two major issues required to be dissolved.

Major concerns:

1.      Tobacco is easy to be transformed to develop stable transgenic plants, so authors should add solid data to improve this manuscript.

2.      Some biochemical evidences required to be added to exhibit the PARylation of protein for strengthening this study.

Author Response

We would like to thank the reviewer for their comments, which I hope, help us to improve the paper.

Major concerns:

  1. Tobacco is easy to be transformed to develop stable transgenic plants, so authors should add solid data to improve this manuscript.

We agree with the reviewer that N. benthamiana can be easily transformed and this will be done in the future.  However, VIGS has been proven to be robust, fast and efficient method  to study functions in plant growth and development, cellular metabolic and signaling pathways, and responses to various biotic and abiotic stresses (https://doi.org/10.3390%2Fijms24065608). PVX vector has been successfully developed and exploited in a variety of loss-of-function experiments in N. benthamiana plants. These explanations have been included in the text. Lines 123-129.

  1. Some biochemical evidences required to be added to exhibit the PARylation of protein for strengthening this study.

Some biochemical data have now been included in the MS. To provide more insights into potential mechanism linking PARP1 activity (PARylation) and tolerance to MV-triggered stress, we have added to the text results showing a possible role of phytohormones such as abscisic acid (ABA) and salicylic acid (SA; free SA and SA glucoside) as intermediates between the PARylation and tolerance (lines 266-295, Figure 4). To validate results on transcriptional control of gene expression of ROS scavenging enzymes, we have also included data on enzymatic activities of two of them, SOD and APX (lines 314-317, Figure 6; see also comments of reviewer 2).

Reviewer 2 Report

Comments and Suggestions for Authors

“Disruption of poly(ADP-ribosyl)ation improves plant tolerance to methyl viologen-mediated oxidative stress via induction of 3 ROS scavenging enzymes” Natalia O. Kalinina, Nadezhda Spechenkova, Irina Ilina,Viktoriya O. Samarskaya, Polina Bagdasarova, Sergey K. Zavriev, Andrew J. Love and Michael Taliansky. In International Journal of Molecular Sciences

Authors want to determine the role(s) of PARP1 and global PARylation in Tobacco leaves submitted to paraquat (methyl viologen). They are also interesting to determine the link(s) of ROS and cell death to a decrease of protein PARylation. They used two different approaches (chemical and genetic) which made the message stronger. 

Thus, this study represents a true interest and may-be an improvement for this agronomic and economical plant. For these reasons, I accepted this paper with few major revisions for publication in IJMS. 

Major comments 

Have the authors a look to PARP2 mRNA regulation in the same type of experiments? According to the experimental result with the KO of PARP1 showing in this work, PARP2 is not be redundant to PARP1 (could thus counter-balance PARP1 KO). What do the authors think about that ? Do they have hypothesis concerning PARP2 that shows the same localization and global activity than PARP1. 

H2O2 is not a ROS but a consequence of ROS production (eg transformation of ROS by peroxidase and SOD) Thus that’s indirect linked to oxidative stress. Could you give an idea of ROS species that you could be involved in this tolerance linked to PARP1 RNAi silencing and did you test it ? 

The same remark for the mRNA expression of the enzymes involved in oxidative stress. Transcriptional level of regulation is often not sufficient to answer to the question. Do the authors monitor to measure their enzymatic activities that must be closed to the reality ? 

Do authors have idea of PARP targets ? 

Minor comments

The first sentence of abstract must be different from the first sentence of introduction that is not the case here. Authors must change

Line 64 Ref 10 instead of Ref 9 ? 

Line105 expression level of PARP1 mRNA instead of production levels

Fig 2B (Line 157) is the PARP1 labelling really into the whole nucleus (line 174)? Do we have one or more nucleoli per nucleus? Could we say that PARP1 is more localized into the Cajal bodies that also makes sense ?  

Author Response

We would like to thank the reviewer for their comments, which I hope, help us to improve the paper.

Major comments

Have the authors a look to PARP2 mRNA regulation in the same type of experiments? According to the experimental result with the KO of PARP1 showing in this work, PARP2 is not be redundant to PARP1 (could thus counter-balance PARP1 KO). What do the authors think about that? Do they have hypothesis concerning PARP2 that shows the same localization and global activity than PARP1.

Our previous data clearly showed that there is PARP1 that plays a key role in regulation of host defences against a virus (Spechenkova et al., 2023). Therefore, in the present study we focused on this particular enzyme to examine its role in responses to abiotic (MV) stress and have shown that its silencing is sufficient to induce some tolerance to MV-mediated stress via induction of ROS scavenging enzymes. However, we still cannot exclude that PARP2 also can contribute to this process and will test this in the future. These remarks have now been included in the text (lines 391-404).

H2O2 is not a ROS but a consequence of ROS production (eg transformation of ROS by peroxidase and SOD) Thus that’s indirect linked to oxidative stress. Could you give an idea of ROS species that you could be involved in this tolerance linked to PARP1 RNAi silencing and did you test it ? 

H2O2 is regarded as a reactive oxygen species (an example of one of many publications illustrating this is doi: 10.3390/ijms20102407), and may also incidentally be derived from other ROS; thus it is a consistent and reliable marker of ROS production. Superoxide anion (O2-) and hydroxy radical (HO) are also produced during ROS production (with H2O2 often as an intermediate between the two), however these are much more reactive than H2O2 and shorter lived. Superoxide anion, hydroxy radical and H2O2 will all likely participate in inducing cellular oxidative damage which is ameliorated via PARP1 RNAi silencing. More clear description has now been included in the text (lines 80-82)

The same remark for the mRNA expression of the enzymes involved in oxidative stress. Transcriptional level of regulation is often not sufficient to answer to the question. Do the authors monitor to measure their enzymatic activities that must be closed to the reality ? 

To validate the RT-PCR data enzymatic activities of two ROS scavenging enzymes (SOD and CAT) were  measured using colorimetric activity kits.  The activity trends of these two enzymes were highly consistent with those from gene expression analyses by RT-PCR (Figure 6; lines 314-317).

Do authors have idea of PARP targets ? 

Our data show that protein PARylation plays important role in plant responses to abiotic stress. Importantly, MV treatment enhances PARylation activity, suggesting that PARylation is an integral part of the regulatory network of stress tolerance in plants. However, PARylation targets and its functions remain largely unknown in plants. We cannot exclude that excessive PAR accumulation induced by MV on its own may merely cause cell death similarly to some mammalian systems (doi: 10.3390/cells8091047), though more specific protein targeting is also possible. For example, taking into account cross talk between PARylation and SA/ABA, we could hypothesize that proteins involved in SA/ABA biosynthesis or signalling may serve as substrates for  PARylation. These remarks have been included in the text (lines 447-452).

Minor comments

The first sentence of abstract must be different from the first sentence of introduction that is not the case here. Authors must change

Has been changed

Line 64 Ref 10 instead of Ref 9 ? 

Both references are given in this line (in new version line 74)

Line105 expression level of PARP1 mRNA instead of production levels

Has been replaced (line 135)

Fig 2B (Line 157) is the PARP1 labelling really into the whole nucleus (line 174)? Do we have one or more nucleoli per nucleus? Could we say that PARP1 is more localized into the Cajal bodies that also makes sense ?  

Normally, each plant nucleus contains a single nucleolus. Under some conditions however, the nucleus may contain more than one nucleoli. In our case all nuclei contained single nucleoli. We agree with the reviewer and have now noted in the text that in Figure 2 PARP1 was located in nuclei and  nucleoplasm (probably in chromatin) preferentially targeting CBs  (lines 224-225)

Reviewer 3 Report

Comments and Suggestions for Authors

In this study, Kalinina et al. reported that repression of PARPs induced the expression of ROS enzymes, thereby improving plant tolerance to oxidative stress. This manuscript is generally well-structured and enhances our understanding of how PARPs are involved in plant tolerance to oxidative stress. Below are my comments and questions on this manuscript.

Line 58: Add a comma to make it more clear, “Plant, like animals, …”.

Line 85-93: This paragraph is highly identical to the Abstract and needs to be modified.

Line 106: The figures should be arranged in the order they appear in the text. The qPCR results should be placed in Figure 1 instead of Figure 2.

Line 197: Is the control solution purely water? More details are needed.

It seems only one VIGS line is included in this study. I wonder if the authors have analyzed more VIGS lines with different repression levels by comparing their response to MV or 3AB treatments.

The statistics in Figure 4 are probably wrong. For example, in the left plot in Panel A, the yellow and green bars under MV treatment are obviously different to the controls, but they are all denoted with “a”, which means no significant difference. Please also check the statistics in other plots and make corrections.

Line 245-254: Please cite relative references.

Line 247: The word “that” appears twice in this sentence. Delete one.

Line 272: The original study of PARP activity in Arabidopsis should be cited besides the two review articles.

Line279-282: Please add references.

Author Response

We would like to thank the reviewer for their comments, which I hope, help us to improve the paper.

Line 58: Add a comma to make it more clear, “Plant, like animals, …”.

Has been added (line 68)

Line 85-93: This paragraph is highly identical to the Abstract and needs to be modified.

                Has now been re-written

Line 106: The figures should be arranged in the order they appear in the text. The qPCR results should be placed in Figure 1 instead of Figure 2.

Have been re-arranged accordingly

Line 197: Is the control solution purely water? More details are needed.

Yes, control solution was water. This has been clarified (lines 151, 489)

It seems only one VIGS line is included in this study. I wonder if the authors have analyzed more VIGS lines with different repression levels by comparing their response to MV or 3AB treatments.

                Results obtained using another line (generated using another PARP1 RNA fragment inserted into a PVX vector) have been added (Figures 1, 3, 4, 5, 6)

The statistics in Figure 4 are probably wrong. For example, in the left plot in Panel A, the yellow and green bars under MV treatment are obviously different to the controls, but they are all denoted with “a”, which means no significant difference. Please also check the statistics in other plots and make corrections.

                Many thanks for this comment: the statistics was wrong indeed. This has now been corrected ( see Figure 5)

Line 245-254: Please cite relative references.

                Have now been cited (lines 344-351)

Line 247: The word “that” appears twice in this sentence. Delete one.

Has now been deleted

Line 272: The original study of PARP activity in Arabidopsis should be cited besides the two review articles.

Has now been added (line 376)

Line279-282: Please add references.

                Has now been added (line 390)

Reviewer 4 Report

Comments and Suggestions for Authors

The study titled "Disruption of poly(ADP-ribosyl)ation improves plant tolerance to methyl viologen-mediated oxidative stress via induction of ROS scavenging enzymes" explores the role of poly(ADP-ribosyl)ation (PARylation) and PARP1 (poly(ADP-ribosyl) polymerase 1) in regulating plant responses to oxidative stress induced by methyl viologen (MV) in Nicotiana benthamiana. The research demonstrates that RNA interference (RNAi) silencing of the PARP1 gene, as well as the pharmacological inhibition of PARP1 using 3-aminobenzamide (3AB), significantly enhance plant tolerance to MV-induced oxidative stress. This study makes a valuable contribution to understanding the role of PARylation in plant stress responses, particularly under conditions of oxidative stress induced by MV. The investigation into PARP1 in Nicotiana benthamiana offers important insights into plant biology, providing detailed mechanistic explanations for how PARP1-mediated PARylation influences plant stress tolerance by regulating the expression of ROS scavenging enzymes. The clear link between PARP1 activity and the regulation of oxidative stress responses contributes to a better understanding of stress tolerance mechanisms in plants. The objectives of this study are well-defined, and the results are sufficiently convincing. While I do not have major concerns, I offer a few suggestions below:

While the study provides important insights into the role of PARP1, it focuses exclusively on this single member of the PARP family, without exploring the potential roles of other PARP family members, such as PARP2 or PARP3. A broader analysis encompassing the entire PARP family could have offered a more comprehensive understanding of the role of PARylation in stress tolerance, and additional introduction or discussion on this point would enhance the clarity of the authors’ perspective.

The study suggests that the increased accumulation of PARylated proteins might result from post-transcriptional activation of PARP1. However, the specific mechanisms underlying this post-transcriptional regulation are not explored. Further discussion on this aspect would be beneficial.

The study does not sufficiently address potential confounding factors that could influence the results. For example, the role of other stress-related pathways, such as those involving abscisic acid (ABA) or salicylic acid (SA), is not thoroughly investigated. These pathways could interact with PARylation, potentially complicating the interpretation of the findings. A more detailed discussion of these interactions would strengthen the study’s conclusions.

The study focuses on short-term responses to oxidative stress but does not consider the potential long-term effects of PARP1 inhibition on plant growth and development. Continuous inhibition of PARP1 might have unintended consequences, such as affecting DNA repair processes or other vital cellular functions, which could compromise plant health in the long term.

Comments on the Quality of English Language

Overall, the manuscript is well-written, with clear and precise language. The authors have effectively communicated complex scientific concepts in a manner that is accessible to readers with a background in molecular biology. However, there are instances where sentences could be further simplified to enhance readability. The grammatical structure of the manuscript is generally strong, with only a few areas that could benefit from revision. The paragraphs are generally well-organised, with each one focused on a specific idea or result. However, some paragraphs could be tightened by removing extraneous information or by better integrating related concepts.

Line 21-22: Change "PARP1 gene and by application of pharmacological inhibitor 3-aminobenzamide (3AB) of PARylation activity." to "PARP1 gene and by applying the pharmacological inhibitor 3-aminobenzamide (3AB) to inhibit PARylation activity."

Line 25-27: Change "This mechanism may be integrated in a specific consolidated network that controls plant sensitivity to oxidative stress by multiple genetically programmed pathways." to "This mechanism may be part of a broader network that regulates plant sensitivity to oxidative stress through various genetically programmed pathways."

Line 83: Change "Reduction in PARP activity has also been shown to facilitate MV-resistance." to "Given the central role of PARP1 in regulating oxidative stress responses, it's important to explore how its reduced activity might contribute to MV-resistance in plants."

Line 85-86: Changr "This study was undertaken to investigate whether and how PARP1 and PARylation regulate responses of Nicotiana benthamiana plants to MV-induced oxidative stress." to "We undertook this study to investigate how PARP1 and PARylation regulate the responses of Nicotiana benthamiana plants to MV-induced oxidative stress."

Line 89-90: Change "In turn, reduced accumulation of PARylated proteins was accompanied with significant increase in gene expression of major ROS scavenging enzymes including CAT (NbCAT2), SOD (NbMnSOD; mitochondrial manganese SOD), APX (NbAPX5) and GR (NbGR) and inhibition of cell death." to "The reduction in PARylated proteins was associated with a significant increase in the expression of major ROS scavenging enzymes, such as CAT (NbCAT2), SOD (NbMnSOD), APX (NbAPX5), and GR (NbGR), which contributed to the inhibition of cell death."

Line 103-104: Change "VIGS constructs were made by insertion of fragments of the PARP1 gene into a potato virus X (PVX) expression vector (pGR106; [24]), giving PVX-PARP1." to "Virus-Induced Gene Silencing (VIGS) constructs were made by inserting fragments of the PARP1 gene into a potato virus X (PVX) expression vector (pGR106; [24]), creating PVX-PARP1."

Line 241-243: Change "Plants are continuously exposed to various types of abiotic (physiological or environmental) stresses that can adversely affect their metabolism, growth and development." to "Plants are constantly exposed to abiotic stresses, both physiological and environmental, that can negatively impact their metabolism, growth, and development."

Line 245-246: Change "Over the past decade, various approaches have been employed to understand the mechanistic molecular basis of plant responses to stress and to identify key genes that that regulate these processes." to "Over the past decade, researchers have employed various approaches to uncover the molecular mechanisms underlying plant stress responses and to identify the key genes that regulate these processes."

Line 273-274: Change "One of the suggested possibilities was that the excessive energy consumption, which is a normally associated with PARP activation under stress conditions was prevented by silencing of PARP." to "One possibility is that silencing PARP prevents the excessive energy consumption typically associated with its activation under stress conditions."

Line 288-289: Change "Collectively, these observations suggest that excessive PARylation may be harmful for plant cells during abiotic (MV-mediated) stress." to  "These findings suggest that excessive PARylation may harm plant cells under MV-mediated stress."

Line 299-300: Change "However, the role (deactivation or suppression of gene expression) of PAR-degrading enzymes such as PARG and NUDIX also cannot be excluded completely." to "However, the potential role of PAR-degrading enzymes like PARG and NUDIX, whether through deactivation or suppression of gene expression, cannot be entirely excluded."

Line 315-317: Change "Spray induced genetic silencing (based on the application of dsRNA) may be more applicable." to "Spray-induced genetic silencing, using the application of dsRNA, may be a more applicable approach."

Author Response

We would like to thank the reviewer for their comments, which I hope, help us to improve the paper.

While I do not have major concerns, I offer a few suggestions below:

While the study provides important insights into the role of PARP1, it focuses exclusively on this single member of the PARP family, without exploring the potential roles of other PARP family members, such as PARP2 or PARP3. A broader analysis encompassing the entire PARP family could have offered a more comprehensive understanding of the role of PARylation in stress tolerance, and additional introduction or discussion on this point would enhance the clarity of the authors’ perspective.

In this paper we studied a role of PARylation in plant responses to MV-mediated oxidative stress. PARylation is catalysed by members of PARP family. Plants encode three PARP proteins but only two of them PARP1 and PARP 2 but not PARP3 possess poly (ADP-ribose) polymerase activity. PARP2 is regarded as predominant PARP enzyme in plant DNA damage and immune responses. However, in our previous study we found that there is PARP1 that plays a key role in regulation of host defences against a virus (Spechenkova et al., 2023). Therefore, here as the first step we extend our previous research by elucidating the role of PARP1 in responses to MV-mediated abiotic stress. The role of the entire PARP family will be investigated in the future. These remarks have been included in the text (lines 391-404)

The study suggests that the increased accumulation of PARylated proteins might result from post-transcriptional activation of PARP1. However, the specific mechanisms underlying this post-transcriptional regulation are not explored. Further discussion on this aspect would be beneficial.

One possibility is that PARP1 can be activated  by single- and double-strand DNA breaks as described for some animal systems  (https://doi.org/10.1038/nsmb.2306). However, other abiotic stress induced factors may also be involved in plants. This has been included in the text (425-428).

The study does not sufficiently address potential confounding factors that could influence the results. For example, the role of other stress-related pathways, such as those involving abscisic acid (ABA) or salicylic acid (SA), is not thoroughly investigated. These pathways could interact with PARylation, potentially complicating the interpretation of the findings. A more detailed discussion of these interactions would strengthen the study’s conclusions.

We have now included in the text data, showing cross-talk between SA/ABA accumulation and PARP1 silencing or inhibition and proposed a model demonstrating the integral connection between poly(ADP-ribosyl)ation activity mediated by PARP1 in response to oxidative stress caused by MV,  changes in production of phytohormones, SA and ABA, induction of ROS scavenging enzymes and stress tolerance (lines 266-295, 436-443, Figure 4)

The study focuses on short-term responses to oxidative stress but does not consider the potential long-term effects of PARP1 inhibition on plant growth and development. Continuous inhibition of PARP1 might have unintended consequences, such as affecting DNA repair processes or other vital cellular functions, which could compromise plant health in the long term.

Such possibility has been mentioned in the text (lines 456-458)

Comments on the Quality of English Language

Overall, the manuscript is well-written, with clear and precise language. The authors have effectively communicated complex scientific concepts in a manner that is accessible to readers with a background in molecular biology. However, there are instances where sentences could be further simplified to enhance readability. The grammatical structure of the manuscript is generally strong, with only a few areas that could benefit from revision. The paragraphs are generally well-organised, with each one focused on a specific idea or result. However, some paragraphs could be tightened by removing extraneous information or by better integrating related concepts.

Line 21-22: Change "PARP1 gene and by application of pharmacological inhibitor 3-aminobenzamide (3AB) of PARylation activity." to "PARP1 gene and by applying the pharmacological inhibitor 3-aminobenzamide (3AB) to inhibit PARylation activity."

Changed (lines 23-24)

Line 25-27: Change "This mechanism may be integrated in a specific consolidated network that controls plant sensitivity to oxidative stress by multiple genetically programmed pathways." to "This mechanism may be part of a broader network that regulates plant sensitivity to oxidative stress through various genetically programmed pathways."

Changed (lines 28-30)

Line 83: Change "Reduction in PARP activity has also been shown to facilitate MV-resistance." to "Given the central role of PARP1 in regulating oxidative stress responses, it's important to explore how its reduced activity might contribute to MV-resistance in plants."

Changed (lines 89-91)

Line 85-86: Changr "This study was undertaken to investigate whether and how PARP1 and PARylation regulate responses of Nicotiana benthamiana plants to MV-induced oxidative stress." to "We undertook this study to investigate how PARP1 and PARylation regulate the responses of Nicotiana benthamiana plants to MV-induced oxidative stress."

Changed (lines 99-100)

Line 89-90: Change "In turn, reduced accumulation of PARylated proteins was accompanied with significant increase in gene expression of major ROS scavenging enzymes including CAT (NbCAT2), SOD (NbMnSOD; mitochondrial manganese SOD), APX (NbAPX5) and GR (NbGR) and inhibition of cell death." to "The reduction in PARylated proteins was associated with a significant increase in the expression of major ROS scavenging enzymes, such as CAT (NbCAT2), SOD (NbMnSOD), APX (NbAPX5), and GR (NbGR), which contributed to the inhibition of cell death."

Changed (lines 105-108)

Line 103-104: Change "VIGS constructs were made by insertion of fragments of the PARP1 gene into a potato virus X (PVX) expression vector (pGR106; [24]), giving PVX-PARP1." to "Virus-Induced Gene Silencing (VIGS) constructs were made by inserting fragments of the PARP1 gene into a potato virus X (PVX) expression vector (pGR106; [24]), creating PVX-PARP1."

Changed (line 131)

Line 241-243: Change "Plants are continuously exposed to various types of abiotic (physiological or environmental) stresses that can adversely affect their metabolism, growth and development." to "Plants are constantly exposed to abiotic stresses, both physiological and environmental, that can negatively impact their metabolism, growth, and development."

Changed (line 336-338)

Line 245-246: Change "Over the past decade, various approaches have been employed to understand the mechanistic molecular basis of plant responses to stress and to identify key genes that that regulate these processes." to "Over the past decade, researchers have employed various approaches to uncover the molecular mechanisms underlying plant stress responses and to identify the key genes that regulate these processes."

Changed (lines 341-343)

Line 273-274: Change "One of the suggested possibilities was that the excessive energy consumption, which is a normally associated with PARP activation under stress conditions was prevented by silencing of PARP." to "One possibility is that silencing PARP prevents the excessive energy consumption typically associated with its activation under stress conditions."

Changed (363-365)

Line 288-289: Change "Collectively, these observations suggest that excessive PARylation may be harmful for plant cells during abiotic (MV-mediated) stress." to  "These findings suggest that excessive PARylation may harm plant cells under MV-mediated stress."

Changed (391-393)

Line 299-300: Change "However, the role (deactivation or suppression of gene expression) of PAR-degrading enzymes such as PARG and NUDIX also cannot be excluded completely." to "However, the potential role of PAR-degrading enzymes like PARG and NUDIX, whether through deactivation or suppression of gene expression, cannot be entirely excluded."

Changed (lines 405-407)

Line 315-317: Change "Spray induced genetic silencing (based on the application of dsRNA) may be more applicable." to "Spray-induced genetic silencing, using the application of dsRNA, may be a more applicable approach."

Changed (lines 446-447)

Round 2

Reviewer 1 Report

Comments and Suggestions for Authors

The revised manuscript improved in quality. And the explanations were provided according to the comments. Now, there is no question to give about this manuscript.

Author Response

We would like to thank the reviewer for positive comments.

Reviewer 3 Report

Comments and Suggestions for Authors

The manuscript is greatly improved, but there are some new issues in the updated version.

Line 203: This sentence is not clear.

Line 269: There is no need to repeat the abbreviations multiple times in the manuscript.

Line 296: Correct the number of the subheading.

Line 317: RT-qPCR?                                                                

Comments on the Quality of English Language

The overall quality of English in this manuscript is good, except for some minor grammatical mistakes.

Line 250: Use “derived from” instead of “derived of”. The same issue in Line 276.

Line 371: Remove “indicate that”.

Line 382: Please change “levels of abscisic acid (ABA)” to abscisic acid (ABA) levels.

Author Response

The manuscript is greatly improved, but there are some new issues in the updated version.

Line 203: This sentence is not clear.

Has now been changed. “distribution” changed for “localisation”

Line 269: There is no need to repeat the abbreviations multiple times in the manuscript.

Abbreviations removed

Line 296: Correct the number of the subheading.

Corrected

Line 317: RT-qPCR?  

Corrected       

The overall quality of English in this manuscript is good, except for some minor grammatical mistakes.

Line 250: Use “derived from” instead of “derived of”. The same issue in Line 276.

Corrected

Line 371: Remove “indicate that”.

Removed

Line 382: Please change “levels of abscisic acid (ABA)” to abscisic acid (ABA) levels.

Changed